# Role of Serotonin Transporter in Eye Development of *Drosophila*
*melanogaster*

**DOI:** 10.3390/ijms21114086

**Published:** 2020-06-08

**Authors:** Tuan L. A. Pham, Tran Duy Binh, Guanchen Liu, Thanh Q. C. Nguyen, Yen D. H. Nguyen, Ritsuko Sahashi, Tran Thanh Men, Kaeko Kamei

**Affiliations:** 1Department of Functional Chemistry, Kyoto Institute of Technology, Kyoto 606-8585, Japan; phamleanhtuan.2807@gmail.com (T.L.A.P.); tdbinh22@gmail.com (T.D.B.); guanchenleo@hotmail.com (G.L.); nqcthanh92@gmail.com (T.Q.C.N.); ndhyen94@gmail.com (Y.D.H.N.); sahashi@bio-energy.jp (R.S.); 2Department of Biology, Cantho University, Cantho 900000, Vietnam; ttmen@ctu.edu.vn

**Keywords:** *Drosophila*, serotonin transporter, eye development, cell death, PI3K/Akt pathway

## Abstract

Serotonin transporter (SerT) in the brain is an important neurotransmitter transporter involved in mental health. However, its role in peripheral organs is poorly understood. In this study, we investigated the function of SerT in the development of the compound eye in *Drosophila melanogaster*. We found that SerT knockdown led to excessive cell death and an increased number of cells in S-phase in the posterior eye imaginal disc. Furthermore, the knockdown of *SerT* in the eye disc suppressed the activation of Akt, and the introduction of *PI3K* effectively rescued this phenotype. These results suggested that SerT plays a role in the healthy eye development of *D.*
*melanogaster* by controlling cell death through the regulation of the PI3K/Akt pathway.

## 1. Introduction

The serotonergic system is conserved from insects to mammals, and it plays a pivotal role in the mental health of organisms [1,2]. The serotonergic system exists in both the brain and peripheral organs. In the brain, serotonin (5-hydroxytryptamine, 5-HT), the core molecule of the system, acts as a neurotransmitter and is associated with the feeling of wellness and happiness [3]. Altered regulation of serotonin concentration in the brain has been recorded not only in a vast range of behavioral functions, such as wake/sleep states, appetite, and sexual behavior, but also in the causation of multiple severe psychotic disorders, such as depression, bipolar disorder, and schizophrenia [4,5]. Although the roles of serotonin in the brain have been intensively investigated, the serotonergic function in peripheral organs, where serotonin acts as a hormonal molecule, is mostly uninvestigated, even though 95% of the serotonin in the whole body is produced by the gastrointestinal tract [6]. One of the essential components in the regulation of serotonin is serotonin transporter (SerT) or 5-hydroxytryptamine transporter (5-HTT), which is a monoamine transporter protein responsible for the reuptake of serotonin into neurons after synaptic transmission in mammals [7]. SerT is a transmembrane transporter that is highly conserved in various organisms from insects to mammals. SerT in the central nervous system is associated with anxiety and depression-related psychological conditions, thereby making it a target for a large number of antidepressant drugs called selective serotonin reuptake inhibitors (SSRIs) [8]. On the other hand, peripheral SerT has been reported to play significant roles in cardiovascular processes, appetite regulation, endocrine mediation, and reproduction [9,10,11,12]. Still, the function of SerT in developmental processes is poorly understood.

*Drosophila melanogaster* is a versatile model organism for researching various biological and physiological mechanisms [13]. The compound eye of an adult fly is composed of approximately 750 unit eyes, known as ommatidia. Each ommatidium consists of a core of eight photoreceptor neurons, R1–R8. *Drosophila* compound eyes are derived from the eye-antennal disc. In the eye disc of *Drosophila* third-instar larva, the morphogenetic furrow (MF) represents the mitotic wave of cells, which sweeps from posterior to anterior. The cells exiting just ahead of MF in the anterior area and within the MF will undergo cell cycle arrest in the synchronized G1 phase, waiting for their fate to be determined, and form a progenitor cell pool. As MF propagates, the progenitor cells are recruited into photoreceptor pre-clusters and specified as several types of photoreceptor neurons (R8, R2, R5, R3, or R4). In cells just at the back of MF in the posterior area, which have not started to differentiate, one more round of division called the second mitotic wave (SMW) occurs to recruit cells into other types of photoreceptor neurons (R1, R6, or R7) [14,15,16]. In the posterior region, after the SMW passes, no additional cell cycles occur within the larval disc. Because there are more cells than needed in the eye disc to form ~750 ommatidia, undifferentiated cells will either later differentiate into photoreceptors or accessory cells, or be removed via apoptosis during pupation [17]. Since the core of photoreceptor neurons is arrayed in a structure that is accurately repetitive in healthy individuals [18], small abnormalities in the compound eyes can be easily observed, which make compound eye tissues ideal for examining cell fate via signaling transduction pathways. Thus, we chose the *Drosophila* compound eye as target tissues for investigating the role of SerT in the eye development.

*Drosophila* and mammals share a highly conserved phosphatidylinositol 3-kinase (PI3K)/Akt signaling pathway, which plays a vital role in regulating cellular growth and energy metabolism [19,20]. Moreover, multiple pieces of evidence showed the relation between this pathway and the development of the *Drosophila* compound eye. Palomero et al. (2007) showed the interaction of PTEN or Akt with Notch results in abnormal eye growth and tumorigenesis [21]. The target of rapamycin (TOR) is also involved in eye development, as TOR directly couples with the adaptor protein Unk. Bateman, J.M. [22] showed that together, TOR and Unk mediate the differentiation fate of photoreceptors R1, R6 and R7. Moreover, the activity of the TOR complexes are required for hyperplasia upon elevated PI3K signaling in the compound eye [23]. Slade and Staveley (2015) showed that numerous novel Akt1 mutants exhibit fewer ommatidia with reduced size [24]. We hypothesized that the PI3K/Akt pathway mediates the role of SerT in the development of the compound eye in *Drosophila melanogaster*.

## 2. Results

### 2.1. SerT Knockdown Disrupts Healthy Eye Development

Despite multiple studies that explore SerT expression in the central nervous system of *Drosophila* [25,26], there is no evidence of SerT expression in the imaginal eye discs. In this research, SerT expression was visualized by a specific antibody for *Drosophila* SerT. Anti-SerT signals were distributed outside of the nucleus throughout the eye imaginal disc, showing high intensity (Figure 1B,B′).

Then, to determine the role of SerT in eye development, we investigated the effect of the reduction of SerT protein in the compound eye by using a GAL4/Upstream activation sequence (GAL4/UAS) system. Two *UAS*-*SerT* RNAi strains that have different inverted repeat sequences downstream of the *UAS* sequence with no overlap were used to exclude the possibility of off-target knockdown. The *UAS*-*SerT* RNAi strains were crossed with the *GMR*-GAL4 driver that expresses GAL4 in the eye under the control of glass enhancer; therefore, the offspring strongly expressed *SerT* RNAi in all the cells behind the MF, causing its specific knockdown in the eye. The knockdown strains, *GMR* > *SerT*-inverted repeat (IR)1 (Figure 1D,D′) and *GMR* > *SerT*-IR2 (Figure 1E,E′), showed a rough phenotype in 100% of the adult compound eyes, in which the ommatidia were oddly shaped, and the bristles were lost. As the *GMR* > *SerT*-IR1 strain expressed a more severe phenotype, we chose this strain for further experiments. These data suggested that SerT played an important role in the development of the compound eye.

### 2.2. SerT Knockdown Induces Cell Death via a Caspase-Dependent Pathway

Previous reports suggested that the rough eye phenotype may be caused by excessive cell death [27,28]. Thus, we stained the eye disc of third-instar larvae with anti-caspase-3 antibody. The number of the caspase-3-positive signal was significantly higher (4.4-fold) in knockdown flies (*SerT*-kd) than in wild-type flies, suggesting an increase in caspase-dependent cell death (Figure 2A,A′). This result was further confirmed by introducing either the *p35* or *diap1* gene, both of which encoded apoptosis inhibitors, into knockdown flies. The flies carrying *UAS-p35* or *UAS-diap1* in the background of *SerT* knockdown showed less a severe phenotype (Figure 2D–D″,E–E″). The ommatidia structures slightly recovered, and the bristles were partially regenerated. Although flies carrying *UAS-gfp* and *SerT*-IR, which were generated as control, showed no significant rescue compared with the knockdown flies (Figure 2C–C″).

### 2.3. SerT Knockdown Increases the Number of Cells in S-Phase in the Eye Imaginal Discs

Given the increased level of cell death, we hypothesized that *SerT*-kd might cause cell cycle defects in the eye imaginal discs. To prove this hypothesis, we utilized the 5-ethynyl-2′deoxyuridine (EdU) incorporation assay, which can detect proliferating S-phase cells. The result showed that there was a significant increase (2.4-fold) in the number of EdU-positive cells in the posterior area of *SerT*-kd flies than in that of control flies (the region surrounded by dotted lines in Figure 3A,A′). Moreover, the inhibition of apoptosis by the expression of DIAP1 significantly decreased the number of S-phase cells (Figure 3A”), suggesting a link between two phenomena. Moreover, we also detected the abnormal organization of cone cell nuclei by DAPI staining, which might be a result of excessive proliferation (Figure 3C,C′). On the other hand, we found that there was no change in the number of cells in the M-phase, as determined by anti-phosphorylated histone 3 antibody, between *SerT*-kd and control flies (Figure 3D,D′,E). This suggests that only the number of cells in S-phase was affected, whereas the process of cell proliferation was not. In addition, we eliminated the possibility of DNA-damage causing excessive proliferation or cell death by staining the posterior eye disc with anti-phosphorylated-gamma-histone 2A, a known DNA-damage marker [29]. Low-signal intensity was observed in both the control and *SerT*-kd (Figure 3F,F′,G), suggesting that DNA-damage did not cause the phenotype.

### 2.4. SerT Knockdown Induces a Rough Eye Phenotype via the PI3K/Akt Pathway

We suspected that the eye phenotype of the *SerT* knockdown fly might be influenced by the PI3k/Akt pathway. This pathway is well known to be involved in the regulation of the cell cycle, and there is also evidence that suggests its participation in the neurogenesis of the compound eye [22]. First, we visualized the Akt activation level using specific *Drosophila* anti-phosphorylated Akt at Ser505 (the equivalent of human Ser473). Figure 4A–D” shows a significant decrease in the level of phosphorylated Akt in the posterior region (the region affected by *SerT*-kd, Figure 4D–D”) of the eye disc of the *SerT*-kd fly compared with that in the control (Figure 4A–B”) and in the anterior region (the region unaffected by *SerT*-kd, Figure 4C–C”) of the same eye disc.

Since Akt activation is regulated via PI3K, we tried to rescue the phenotype by introducing *PI3K* in *SerT*-kd flies. The result showed the significant restoration of ommatidia shape and bristle formation (Figure 4G–G”), compared with those in the control flies carrying *gfp* (Figure 4F–F”). A representative image of the compound eyes of flies with *PI3K* overexpression, but without *SerT*-RNAi, is also shown (Figure 4H–H”). Taken together, the results indicated that the levels of Akt phosphorylation were reduced by *SerT* knockdown. The levels of phosphorylated Akt are negatively regulated by the feedback of Akt itself, and it is often difficult to interpret changes in Akt phosphorylation. This observation alone is not proof of reduced PI3K/Akt activity, but it is suggestive.

## 3. Discussion

SerT plays important roles in various processes in the peripheral system [30]. In this study, we visualized the expression of SerT in *Drosophila* imaginal disc by specific antibody (Figure 1B,B′), which led us to elucidate the function of SerT in eye development. The effect of the knockdown of the *SerT* gene using a *GMR*-GAL4 driver resulted in a noticeable rough eye phenotype, represented by the abnormal organization of ommatidia and the complete loss of bristles (Figure 1D,D′,E,E′). Previous studies indicated that the rough eye phenotype was possibly caused by the disruption of the cell cycle in the development of the compound eyes, which may lead to the premature termination of cells via a caspase-dependent pathway [27,31]. Our results indeed showed an increase in caspase-dependent cell death in knockdown flies, which was effectively rescued by the caspase inhibitor DIAP1 or p35 (Figure 2D–D”,E–E”). Figure 3A′ also showed that, under *SerT*-kd, the number of cells in the S-phase accumulated in the posterior region was significantly higher in knockdown flies than in the control, though the first mitotic wave (indicated by MF) and the second wave had passed.

Although a large number of cells were in S-phase, pH3S10 staining showed no change in the number of cells undergoing mitosis between *SerT*-kd and the controls. A previous study showed that *rux* mutation caused defects in the cell cycle regulation that led to the premature entry into S phase (represented by substantial increases in 5-bromo-2’-deoxyuridine staining), resulting in a similar but more severe rough eye phenotype to that observed in the present study and in the absence of changes in mitosis [32]. Moreover, our recent study of the regulation of the developing wing suggested that the knockdown of dLipin by an *sd*-GAL4 driver induces the accumulation of cells in S-phase while also decreasing the number of mitotic cells [33]. These studies suggest that increases in S- and M-phase cells can be regulated differently due to cell cycle defects; however, the underlying mechanism needs to be investigated further. The result shown in Figure 3F,F′, describing a lack of DNA damage, excludes the involvement of the DNA-repair mechanism in the phenotype. An alternative hypothesis might be that the excess numbers of S-phase cells might be induced by compensatory mechanisms related to cell death. Although this is suggestive, key evidence is missing, especially the cause of differential regulation of the mitotic phase. The abnormal organization of cone cell nuclei (Figure 3C′) suggested that excessive S-phase proliferation may cause the elevated recruitment of DNA in this cell type, strengthening our claim in the link between S-phase proliferation and rough eye phenotype. However, whether extra cone cells are formed or not is unclear and requires further investigation.

Then, our results suggested that the excessive caspase-dependent cell death in the posterior region of the eye disc was induced by the decreased level of the PI3K/Akt pathway activation. Clinical and experimental data highlighted the insulin-induced PI3K/Akt pathway, a universal pathway in yeast, insects, mice, and mammals, as a common pathway in the stabilization of cell growth under the control of nutrition [34]. The PI3K/Akt pathway is widely known as a crucial inhibitor of apoptotic effectors in the growth-signaling pathway. The activation of this pathway can reduce apoptosis in various types of cells, including neuronal, ischemia-inducing myocardial, and tumor cells [35,36,37,38]. In the rat hippocampus, the PI3K/Akt signaling pathway plays a pivotal role in neuronal apoptosis after inducing subarachnoid hemorrhage [39]. Furthermore, the inhibition of the PI3K/Akt pathway promotes the activation of caspase-3, subsequently increasing apoptotic cell death in diabetic rats [40]. Moreover, the activation of PI3K/Akt is effectively controlled by the insulin network [41,42]. The inhibition of SerT via a genetic or pharmacological mechanism results in insulin resistance prior to adiposity [43,44,45]. Previous research revealed that in SerT-deficient mice, JNK activity is exalted, and insulin-induced Akt activation is declined, whereas the elevation of AKT signaling by PTEN deficiency rescues the glucose tolerance phenotype. This study also claimed that SerT-deficiency downregulates insulin action and is responsible for impaired PI3K/Akt signaling in the peripheral system [43]. Consistent with these previous findings, our study also showed a link between SerT and PI3K/Akt signaling.

SerT functions are tightly connected to serotonin. SerT knockdown will diminish the clearance of excessive serotonin, which may, in turn, elevates the serotonin accumulation in cells. In mammals, serotonin induces the synthesis and release of insulin, as well as enhances the sensitivity of its target tissue [46]. In *Drosophila*, serotonergic neurons express a GTPase NS3, which controls growth via insulin signaling. Furthermore, Nässel et al. provided a more detailed mechanism that one of the serotonin receptors, 5-HT1A, inhibits adenylate cyclase and protein kinase A, thus inactivating cAMP response element-binding protein, which in turn stimulates insulin signaling [47]. Multiple studies showed that increasing serotonin levels activates the PI3K/Akt pathway in cancer cells and neurodegenerative Parkinson’s disease cellular model [48,49]. This discrepancy can be explained by the different serotonin regulatory processes in serotonergic network interactions. The serotonergic system is complex and regulated by multiple factors, and the regulatory feedback caused by excessive serotonin levels have been previously recorded [50,51]. In order to clarify the detailed interaction within the serotonergic system, the assay of serotonin levels in *SerT* knockdown flies is further needed. Further studies to reveal the specific cell types affected by SerT during eye disc development are warranted.

In conclusion, the cell fate of *Drosophila* imaginal eye disc is strictly regulated by a series of events, and numerous molecules have been found to control these events. In this study, we showed that the cell death induced by the suppressed activation of the PI3K/Akt pathway resulted in a rough eye phenotype in *SerT*-kd flies. Thus, we concluded that SerT played a role in normal eye development by controlling caspase-dependent cell death through the PI3K/Akt pathway. The detailed mechanism of the link between SerT and this cascade requires further investigations.

## 4. Materials and Methods

### 4.1. Fly Stocks

Fly stocks were maintained at 25 °C on standard food containing 0.65% agar, 10% glucose, 4% dry yeast, and 5% cornmeal. Transgenic flies with UAS-*dSerT*-IR_686–1079_ (*SerT*-IR1) and UAS-*dSerT*-IR_1740–1760_ (*SerT*-IR2) were obtained from the Vienna *Drosophila* Resource Center (#100584; Vienna, Austria) and Bloomington *Drosophila* Stock Center (#62985; Bloomington, IN, USA), respectively. These flies carried an IR of the *SerT* gene (targeting regions from nucleotide 686 to 1079 and from 1740 to 1760, respectively) downstream of the UAS sequence, on the second chromosome. All other flies used in this study were obtained from Bloomington *Drosophila* Stock Center: UAS-*gfp* (#1522), UAS-*p35* (#5072), UAS-*diap1* (#6657), UAS-*PI3K* (#8287). *yw* flies were used as the wild-type strain.

#### Scanning Electron Microscopy

All flies were anesthetized by CO_2_ and mounted on a holder. A VE-7800 scanning electron microscope (Keyence, Osaka, Japan) was used to observe the compound eyes of adult flies. At least five adult male flies were observed in each experiment.

### 4.2. Immunostaining

Cells in the S-phase were detected using Click-iT EdU (5-ethynyl-2′-deoxyuridine) labeling Alexa Fluor 594 Imaging Kit (Invitrogen, Carlsbad, CA, USA) [33]. Third instar larvae were dissected in phosphate buffer saline (PBS), and the eye discs were fixed in 4% paraformaldehyde for 20 min at 25 °C. After washing with PBS containing 0.3% Triton X-100 (PBST), the samples were blocked with 0.1% PBST and 10% normal goat serum for 30 min at 25 °C, and incubated with diluted primary antibodies in 0.1% PBST and 10% normal goat serum for 16 h at 4 °C [52,53]. The following antibodies were used as primary antibodies: rabbit anti-*Drosophila* SerT antibody (1:200; S1001-25H, USBio, Salem, MA, USA), rabbit anti-cleaved caspase-3 antibody (1:500; Sigma-Aldrich, St. Louis, MO, USA), anti-Phospho-Histone H3 (Ser10) (D2C8) XP Rabbit mAb conjugated with Alexa488 (1:400; Cell Signaling Technology, Danvers, MA, USA), anti-phosphorylated-histone 2A gamma variant antibody (1:400; DSHB, Iowa, IA, USA), and anti-phosphorylated-*Drosophila* Akt (Ser505) antibody (1:200; Cell Signaling Technology, Danvers, MA, USA). After washing with 0.3% PBST, samples were incubated with secondary antibodies labeled with either Alexa488 or Alexa 594 (Goat anti-rabbit IgG 1:1000 and Goat anti-mouse IgG 1:800; Abcam, Cambridge, UK) for 2 h at 25 °C. After further washing with 0.1% PBST, samples were mounted in a Vectashield mounting medium (Vector Laboratories, Burlingame, CA, USA) and observed by using a Fluoview Fv10i-0 confocal laser scanning microscope (Olympus, Tokyo, Japan). The signals were analyzed by the MetaMorph software (Molecular Devices, Sunnyvale, CA, USA).

### 4.3. Statistical Analysis

Signals in the posterior region of the MF were counted from at least six eye imaginal discs. All experiments were repeated at least three times. Statistical analyses were performed using the Student’s *t*-test and one-way ANOVA. Error bars represent the standard error of the mean (SEM), and all the data are shown as means ± SEM. Differences with *p*-values of <0.05 were considered significant.

## Figures and Tables

**Figure 1 ijms-21-04086-f001:**
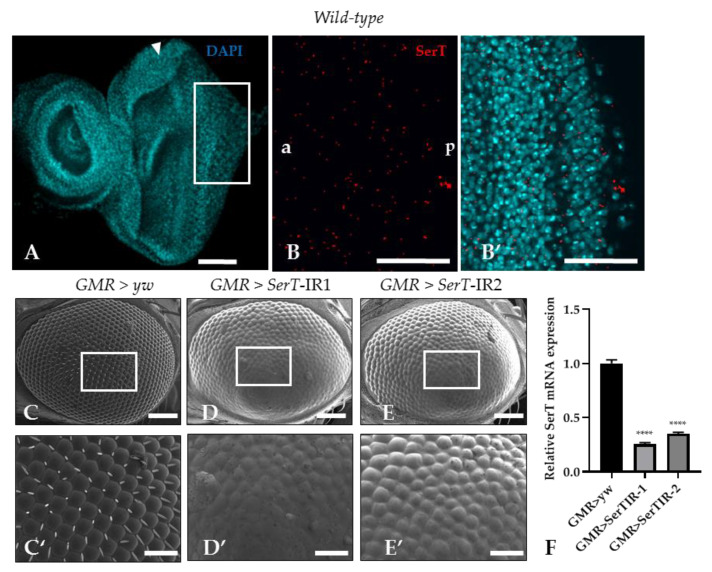
*SerT* knockdown induces a rough phenotype in the adult compound eye. The eye imaginal discs of wild-type third instar larvae were stained with 4′,6-diamidino-2-phenylindole (DAPI) to visualize the nucleus (**A**). The posterior region in the box stained with rabbit anti-*Drosophila* SerT antibody followed by anti-rabbit IgG Alexa Fluor^TM^ 594 antibody was used for detecting SerT (**B**), and the merged image is shown (**B′**). The images are representative of 10 eye imaginal discs. Scanning electron micrographs of the compound eyes of flies carrying *GMR*-Gal4/*yw*; +; + (**C**), *GM*R-Gal4/Y; UAS-*SerT*-IR1/+; +, (**D**), *GMR*-Gal4/Y; and UAS-*SerT*-IR2/+; + (**E**). Larger images of the boxed regions are also shown (**C′**–**E′**). Phenotypes were observed independently in at least three individuals of each fly lines, and no significant change was found in three individuals of the same line. Relative mRNA expression of *SerT* gene in eye discs of flies contains *GMR* > *yw*, *GMR* > *SerT*IR-1, and *GMR* > *SerT*IR-2, *n* = 5 (**F**). Scale bars indicate 100 μm (**A**,**B**,**B′**,**C**–**E**) and 30 μm (**C′**–**E′**). The triangles indicate MF. a, anterior; p, posterior; **** *p* < 0.0001

**Figure 2 ijms-21-04086-f002:**
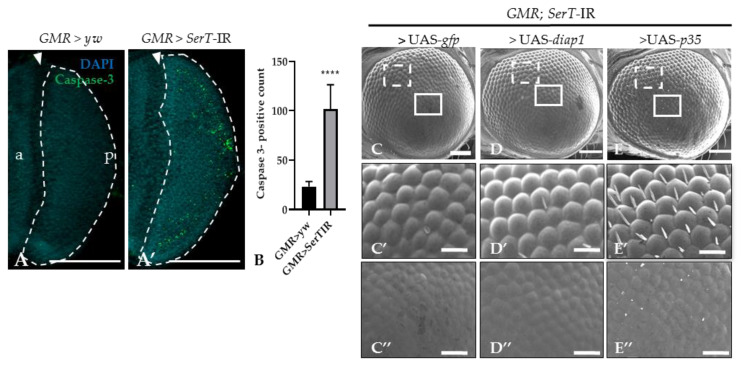
*SerT* knockdown induces caspase-dependent cell death. Region posterior to the morphogenetic furrow (MF) in the eye discs of third-instar larvae carrying *GMR > yw* as a control (**A**) and *GMR > SerT-IR* as *SerT*-kd (**A′**) were stained with anti-caspase-3 antibody followed by secondary antibodies labeled with Alexa488. Quantification of caspase-3 positive signal count in the posterior region surrounded by dotted lines (**B**). Scanning electron micrograph of adult compound eyes of flies carrying *GMR-Gal4/Y, SerT-IR/*UAS*-gfp, +* (**C**), *GMR-Gal4/Y, SerT-IR/*UAS*-diap1*, + (**D**), and *GMR-Gal4/Y, SerT-IR/*UAS*-p35, +* (**E**). Larger images of the boxed regions with dotted white line and solid white line in (**C**–**E**) are shown in (**C′**–**E′**,**C″**–**E″)**, respectively. Scale bar indicates 100 (**A**,**A’**,**C**–**E**) and 30 μm (**C′**–**E′**,**C″**–**E″**). The triangles point to the MF. a, anterior; p, posterior; ****, *p* < 0.0001.

**Figure 3 ijms-21-04086-f003:**
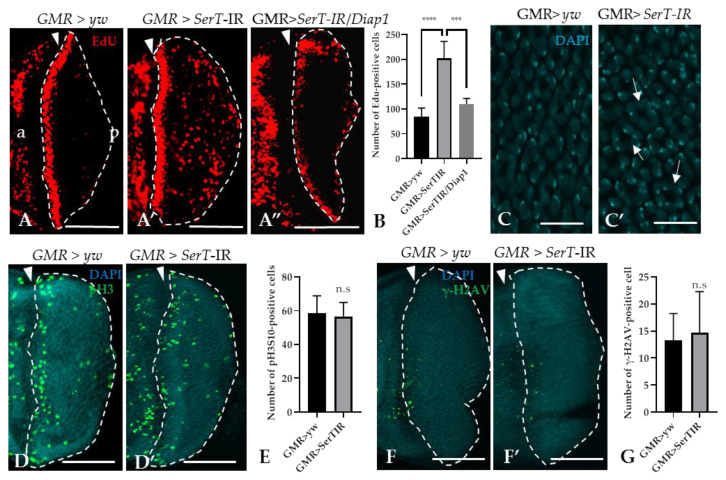
Knockdown of *SerT* induces a high accumulation of cells in the S-phase, but not in the M-phase. The region posterior to the MF stained with EdU in third instar larvae carrying *GMR* > *yw* (**A**), *GMR* > *SerT*-IR (**A′**), and *GMR* > *SerT*-IR/*diap1* (**A”**). Quantification of EdU-positive cells in the posterior region surrounded by dotted lines (**B**). DAPI staining of the nuclei of cone cells of third-instar larvae carrying *GMR* > *yw* (**C**) and *GM*R > *SerT*-IR (**C′**). Arrows point to the abnormal formation of the nuclei. Posterior region of third-instar larvae carrying *GMR* > *yw* and *GMR* > *SerT*-IR, respectively, were stained with anti-phosphorylated-histone 3 (Ser10) antibody conjugated with Alexa488 (**D**,**D′**) and with anti-phosphorylated-gamma-Histone 2A antibody (**F**,**F′**). Quantification of pH3S10-positive cells and γ-H2AV-positive cells in the region surrounded by dotted lines, respectively (**E**,**G**). Scale bars indicate 100 μm (**A**–**A”**,**D**,**D′**,**F**,**F′**) and 5 μm (**C**,**C′**). The triangles indicate the MF. a, anterior; p, posterior. ****, *p* < 0.0001; ***, *p* < 0.001; n.s., not significant.

**Figure 4 ijms-21-04086-f004:**
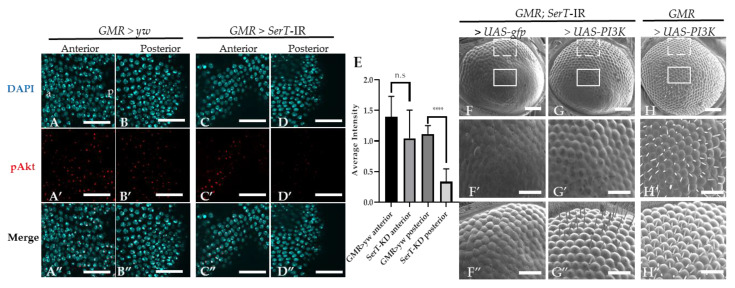
Knockdown of *SerT* reduces the levels of phosphorylated Akt protein. The anterior and posterior region to the MF in the eye discs of third-instar larvae carrying GMR > yw as a control (**A**–**A”**,**B**–**B”**, respectively) and GMR > *SerT*-IR as *SerT*-kd (**C**–**C”**,**D**–**D”**, respectively) were stained with DAPI (blue) and anti-phosphorylated-Akt antibody followed by secondary antibodies labeled with Alexa594 (red). Merged images are also shown (**A”**,**B”**,**C”**,**D”**). The images are representative of 5 eye imaginal discs. Quantification of the phosphorylated Akt average intensity (**E**). Scanning electron micrograph of the adult compound eyes of flies carrying *GMR*-Gal4/Y, *SerT*-IR/UAS-*gfp*, + (**F**), *GMR*-Gal4/Y, *SerT*-IR/UAS-*PI3K*, + (**G**), and *GMR*-Gal4/Y, UAS-*PI3K*, + (**H**). Larger images of regions surrounded by box and dotted lines are shown (**F′**–**H′**,**F”**–**H”**, respectively). Scale bar indicates 15 μm (**A**–**D”**), 100 μm (**F**–**H**) and 30 μm (**F′**–**H”**). a, anterior; p, posterior; ****, *p* < 0.0001.

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
