# Peer review of "Role of Serotonin Transporter in Eye Development of *Drosophila"

_ijms, 2020, doi:10.3390/ijms21114086_

Round 1

Reviewer 1 Report

Serotonin is one of the neurotransmitters in the central nervous system that carries signals between neurons and it also acts as a hormone. The serotonin transporter has been involved in mental health. In this study, Pham and colleagues study the role of SerT in the peripheral tissue, the developing eye of Drosophila melanogaster.

The loss of SerT leads to apoptosis coupled with compensatory proliferation. How it does so is unclear because the authors overexpress SerT-RNAI transgenes in all cells behind the morphogenetic furrow and thus it is not clear why some cells die and other cells proliferate. It could be both that SerT KD promote excess proliferation but cells do not undergo the complete cell cycle and undergo apoptosis.

The authors show that the reduction of SerT in the developing eye is accompanied by a reduced expression of phosphorylated TOR and claim that the overexpression of PI3K transgene effectively rescued this phenotype. Unfortunately, the images in the manuscript do not support such an "effective" rescue, as only the cells in the anterior part of the eye show bristles, but the eye in general is abnormally shaped and still rough. In summary, the article describes that the external appearance of the eye is rough when SerT RNAi transgenes are expressed under the control of GMR-Gal4 line, but does not describe the cellular basis for this defect and adds a series of analyzes that seem randomly selected to try to link the effect of rough eye with apoptosis and / or alterations in cell proliferation and PI3K signaling. The result is not very encouraging and it does not advance much the current state of the field. It does not clarify where the serotonin that would accumulate in extracellular space comes from and how its increase would affect (inhibit) TOR or the PI3K pathway since SerT/5-HT is normally related to the production and release of the insulin/insulin-like peptides.

Specific comments

  1. 95% of serotonin is produced by the gastrointestinal tract (Terry, Margolies 2017)
  2. An essential component in the regulation of serotonin is the enzyme Trh—tryptophan hydrolase in addition to the SerT. SerT is important for the reuptake of serotonin into neurons after synaptic transmission in mammals. Inhibitors of SerT called selective serotonin-reuptake inhibitors (SSRIs). These inhibitors are the most common used antidepressants. They act by increasing the levels the levels of serotonin. The inhibitors may cause a high level of serotonin to accumulate in the body, leading to the so-called serotonin syndrome.
  3. Page 66-68. The authors hypothesized that the PI3K/AKT/TPR is essential for normal development of Drosophila compound eye. Thus, they study the role of SerT in the development of the compound eye in Drosophila melanogaster via the PI3K/AKT/TOR pathway.

There have been many studies of PI3K/AKT in the developing eye using clones of mutant Pi3K Akt or other pathway components in the developing eye which should be cited the authors are ignoring (Verdu et al., 1998; Staveley et al 1998, Palomero et al 2007, Hietakangas and Cohen, 2007; Chuang et al. 2014, Slade and Staveley, 2015) etc. .

  1. Figure 1B. One cannot see any SerT expression with exception of a few dots in the periphery of the eye imaginal discs hence outside the eye proper. It is not clear whether this antibody reflect endogenous expression as judged by this image.
  2. GMR>SerT-IR1 and GMR>SerT-IR2. A qRT-PCR assay of the efficiency of the SerT-IRs should be included.
  3. The authors state “these data suggested that SerT played an essential role in the development of the compound eye”

Essential means indispensable, vital. Since the phenotype caused by the SerT-RNAi is neither the lack of the eye nor the lethality of the animal, the qualification as an essential role seems as an overstatement.

  1. Dysfunction seems to suggest that some part of SerT doesn’t work well. However, what the authors show is the knockdown of SerT levels and this is what they should state in the subtitle
  2. The external rough eye analysis needs to be complemented with analysis of section of adult eyes to evaluate what cells are affected, or missing.
  3. Different alterations can cause a rough phenotype, in addition to excess in apoptosis. For example, mutants that cause excess or reduction of photoreceptor cells due to problems in recruiting photoreceptor cells or defects in the second wave of mitosis or a loss of apoptosis can all cause a rough eye.
  4. Figure 2D-D '. The total size of the eye is manifestly larger in GMR>SerT-RNAi>Diap1 than that of GMR> SerT-RNAi. However, this increase in size is not seen in p35. This must be quantified and explained.
  5. Overexpression of Diap1 and p35 does not rescue completely or obviously the rough eye.
    It is essential that all these phenotypes are analyzed in tangential sections of the adult eye. Phalloidin or other staining of pupal eyes is also needed along with cell death assays in pupal development to determine which cell types of the ommatids are affected and which part of the phenotype is or is not rescued by the expression of antiapoptotic factors. Also, it must be noted that overexpression of the antiapoptotic factors using GMR-Gal4 interferes with endogenous PCD, making it necessary the analysis of individual GMR> Diap1 and GMR> P35 without SerT.
  6. Caspase staining in Figure 2A should be complemented with co-staining with pan-neural marker Elav or other and cone cell marker Cut. BarH1 is also a marker of pigment cells and could be used to determine with cell type is sensitive to SerT knockdown.
  7. EdU staining shows a clear deregulation of S phase posterior to the morphogenetic furrow. How is that this does not alter the number of mitotic cells? Figure 3C suggests that there is an alteration of mitosis. In particular, some posterior cells that would normally be out of cycle seem to keep dividing. The important thing is to determine which cells are affected. And to explain how apoptosis induce compensatory proliferation without changes in the number of PH3 positive cells?

Does SerT-RNAi affect the recruitment of photoreceptor cells? If so, which ones? R8, R1, R5? or R1 R6 and R7? What are the cells that continue dividing? Are these the post mitotic PR cells which are dying ? Or the uncommitted retinal cells? What relationship is there between the PR cells, the cells that express caspase and those that are positive for EdU?

  1. The Pi3K/AKT/TOR has important roles in growth and energy metabolism but neurogenesis is not a ‘well-known’ role of this pathway.
  2. The Figure 4 shows a rather scattered expression of dTOR? And, more importantly, the expression of phospho-TOR is eliminated both in the anterior region and in the posterior region after the morphogenetic furrow in GMR> SerT-RNAi. This must be explained.
  3. Figure 4D-D '. Manipulation of SerT with RNAi and GMR causes a rough eye phenotype in which the posterior part of the eye is always more affected than the anterior part. In some of the figures, the authors show both areas, mainly to illustrate the rough eye defect. However, for the PI3K "rescue", the authors only show the anterior part (the one that is always least affected) while for the GFP control they show the center that is normally more affected. In addition, control of the effect of Pi3K overexpression without SerT-RNAi must be included.
  4. This image only shows that there is some rescue of the inter-ommatidial bristles of the anterior region of the eye in the condition GMR> SerT-RNAi> PI3K. Since the bristles develop during pupal development and the caspase, EdU and dTOR staining are done in the eye disc during the larval stage when recruitment of PR cells and cone cells takes place, and regulation of the second mitotic division, the rescue of the bristles seems irrelevant to the ‘early’ basis of the rough eye defect. The authors must analyze the basis of the rough eye and not simply look at the external part of the eye and make overinterpretation of what might or might not be affected.

If SerT is required in the eye by raising serotonin levels which is the leading cause of the defects in this peripheral tissue, the authors should provide evidence of a role of serotonin through its receptor in the developing eye.

A key missing experiment is to link SerT to serotonin signaling in the normal eye development.

  1. “SerT acts as a hormonal mediator”

       SerT is a serotonin transporter and thus it is not a hormonal mediator.

  1. The authors cannot claim excess proliferation if they found no changes in PH3 positive cells.

21. There is increased caspase staining but this is neither “excessive” nor is it a demonstration that the cells are dying. To demonstrate cell deaths, other tests such as Tunnel must be done. 22. Page 211. The authors show a decline in the phosphoTOR staining. Given that PI3K/AKT overexpression could also promote cell proliferation the mechanism of ‘rescue’ is unclear.As discussed above, the authors have not clarify which is the primary defect as both cells that would increase caspase and those that would increase EdU are expected to express the SerT-RNAi.23. Please correct the citation —Reference 47 which is not related to cancer.  

In summary, the authors need to fully clarify what is the primary effect of SerT knockdown in the developing eye and present experiments that directly address  the cause and the mechanism of the external " rough eye".

Author Response

First of all, thanks for your valuable comments. We understand your concerns regarding our manuscript and the academic level may not match your requirements. However, we want to highlight that, in this study, we aimed to evaluate the phenotypes associated with SerT-KD during eye development. We identified a basic, functional link between SerT, PI3K pathway, and developing eye. Investigation of the specific cell type that is affected, and the exact cellular defect are not included in this manuscript. 

Please see the attachment for point-to-point response, and kindly note that all revisions to the manuscript are highlighted in yellow.

Reviewer 2 Report

The present manuscript by Pham and colleagues titled ‘Role of Serotonin Transporter in Eye Development of Drosophila melanogaster’ focuses on possible role of serotonin transporter (SerT) during the eye development. Authors report that the knockdown of SerT in GMR GAL4 driven manner results in a rough eye phenotype. Authors attempt to address the mechanism behind the rough eye phenotype. Firstly, authors check for apoptotic activity using cleaved caspase-3 antibody and report increased caspase activity in SerT knockdown flies. Secondly, authors report that SerT knockdown induces cell proliferation using 5-ethynyl-2’deoxyuridine (EdU) incorporation assay. Authors then claim that the PI3K/Akt/TOR pathway is involved in SERT knockdown mediated rough eye phenotype using anti-phosphorylated TOR (at Ser2448) antibody. Over-expression of PI3K in SerT knockdown background considerably rescues the rough eye phenotype. Authors conclude that knockdown of SerT suppresses PI3K/Akt/TOR pathway thereby affecting the normal eye development.

I have following major concerns about the current work

  1. Fig 1B’, authors use mouse anti-Drosophila SerT antibody (1:200; USBio, Massachusetts, USA). I would encourage authors to show a western blot defining the specificity of the antibody. All mouse anti-SerT antibodies on USbio lab catalog are useful against Amino (N)-terminus of the rodent SerT. The N-terminus of Drosophila and rat/mouse vary significantly. Instead, authors could use Bloomington stock number 60529 which labels the truncated version of GFP tagged-SerT.
  2. The sequence alignment shows that Drosophila TOR has proline (amino acid 2369) instead of serine at human TOR 2448 equivalent position. The phosphorylation at this position looks unlikely. Over-expression of PI3K in SerT knockdown background (Fig. 4 D and D’) does seems to restore the ommatidia shape, the bristles however, look longer. Authors could overexpress TOR, AKT and PTEN in SerT knockdown background to confirm the interaction of SERT and PI3K/Akt/TOR pathway.
  3. I would advice the authors to not use the term ‘wild-type allele’ if GAL4-UAS system is used to drive the overexpression of PI3K, diap1 and p35. Wild-type allele is typically denoted as ‘+’ meaning the naturally occurring allele. In the current context wildtype allele will mean the gene is expressed from endogenous promoter.

The manuscript needs to be rewritten to account for above-mentioned points.

Minor comments or suggestions:

  1. Authors could write UAS-p35 instead of p35-OV as this confuses the readers. Alternatively, authors could mention the Bloomington stock number and then call it p35-OV.
  2. Authors could better enlist the flies used in the assays, preferably as a table. Bloomington stock center provide multiple lines for same gene.
  3. Authors could correct several typos in the manuscript (e.g. Figure 5).  
  4. Did authors observed change in the adult eye size, if so, it would be worth mentioning.

Author Response

Comment 1. Fig 1B’, authors use mouse anti-Drosophila SerT antibody (1:200; USBio, Massachusetts, USA). I would encourage authors to show a western blot defining the specificity of the antibody. All mouse anti-SerT antibodies on USbio lab catalog are useful against Amino (N)-terminus of the rodent SerT. The N-terminus of Drosophila and rat/mouse vary significantly. Instead, authors could use Bloomington stock number 60529 which labels the truncated version of GFP tagged-SerT.

>Response: Thank you for your suggestion. We apologize for the misunderstanding. We did not use mouse anti-SerT antibody. We used rabbit anti-Drosophila SerT antibody (S1001-25H, USBio), which is specific for Drosophila SerT. We also performed western blotting for antibody specificity (Fig. 1C). We supplemented the description in Materials and Methods (Line 264)

Comment 2. The sequence alignment shows that Drosophila TOR has proline (amino acid 2369) instead of serine at human TOR 2448 equivalent position. The phosphorylation at this position looks unlikely. Over-expression of PI3K in SerT knockdown background (Fig. 4 D and D’) does seems to restore the ommatidia shape, the bristles however, look longer. Authors could overexpress TOR, AKT and PTEN in SerT knockdown background to confirm the interaction of SERT and PI3K/Akt/TOR pathway.

>Response: Thank you for your comment. We agree that the detection of TOR phosphorylation at Ser2448 is not adequate for Drosophila. We replaced this experiment with the pAkt staining, which is a component of the PI3K pathway, and rewrote the manuscript accordingly (Fig. 4A-D”, line 157-161).

The bristles of PI3K rescue eyes are indeed longer. We also provided an additional figure of PI3K overexpression without SerT-KD (Fig. 4H-H”). Thank you for the valuable suggestions. Although we would like to follow your suggestions for our upcoming investigation, unfortunately, due to limited time, we could not include those experiments in this manuscript.

Comment 3. I would advice the authors to not use the term ‘wild-type allele’ if GAL4-UAS system is used to drive the overexpression of PI3K, diap1 and p35. Wild-type allele is typically denoted as ‘+’ meaning the naturally occurring allele. In the current context wildtype allele will mean the gene is expressed from endogenous promoter.

> Response: Thank you for your suggestion. We corrected the manuscript (line 109, 190).

Minor comments or suggestions:

Comment 1. Authors could write UAS-p35 instead of p35-OV as this confuses the readers. Alternatively, authors could mention the Bloomington stock number and then call it p35-OV.

> Response: Thank you. We revised all “gene-OV” to “UAS-gene” (line 110, 112, Captions of Figure 2 and Figure 4).

Comment 2. Authors could better enlist the flies used in the assays, preferably as a table. Bloomington stock center provide multiple lines for same gene.

> Response: We added the list of used flies (line 250-252).

Comment 3. Authors could correct several typos in the manuscript (e.g. Figure 5).

> Response: We corrected the manuscript (Line 175, 177).

Comment 4. Did authors observed change in the adult eye size, if so, it would be worth mentioning.

> Response: We observed the adult eye size and found no significant change. We rescale all images to avoid distorsion. All images have been normalized so that no eye “seems” larger. (Figure 1D-F).

Reviewer 3 Report

The manuscript by Tuan Pham and colleagues describes the role that Serotonin transporter (SerT) plays in the developing compound eye of the fruit fly, Drosophila melanogaster. The authors show that SerT is expressed in the eye, that RNAi based knock-downs results in eye defects, that apoptosis is increased, and that apoptosis induced proliferation (AIP) is induced. These represent the first description that SerT plays a role in the eye.

All of the data presented in the paper is clear and support the conclusions that are made by the authors. I also like the logical presentation of the experiments. The images are off generally high quality and the writing is very good. It was an interesting paper to read.

I do have a couple of suggestions that I hope the authors will consider.

  1. In panel IB, the SerT signal in most of the cells is very hard to see. I had to blow up the image on my computer and increase the brightness of my screen quite a bit. It would be really helpful if the the authors could provide a better image that shows SerT expression.
  2. I think the manuscript could really benefit from a more detailed examination of the defects in eye development. It would be really nice to see what ae the specific effects on photoreceptor, cone, and pigment cells fate specification in SerT mutants. I would suggest including one or two new figures on phenotypes in larval and pupal discs as well as sections of adult retinas.
  3. The “bald” phenotype (complete loss of eye bristles) is seen when wingless is overexpressed using the sevenless enhancer. That enhancer is easily available from several laboratories and also exists as a GAL4 line. Maybe the authors could use this instead of the GMR GAL4 driver to limit the cells in which the SerT is reduced. It would be interesting if the loss of bristles could be recapitulated by removing SerT in just the sevenless expressing cells.

Author Response

Response to Reviewer 3’s comments.

Comment 1. In panel IB, the SerT signal in most of the cells is very hard to see. I had to blow up the image on my computer and increase the brightness of my screen quite a bit. It would be really helpful if the the authors could provide a better image that shows SerT expression.

>Response: We are sorry for the inconvenience. We provided a clearer image with a higher resolution (Figure 1B).

Comment 2. I think the manuscript could really benefit from a more detailed examination of the defects in eye development. It would be really nice to see what ae the specific effects on photoreceptor, cone, and pigment cells fate specification in SerT mutants. I would suggest including one or two new figures on phenotypes in larval and pupal discs as well as sections of adult retinas.

>Response: Thank you for your valuable suggestions. The specific phenotypic effects of SerT-KD during development is our next goal. Due to limited time, we apologize for the inability to perform the suggested experiments in this manuscript. We would like to use your suggestions for upcoming studies.

Comment 3. The “bald” phenotype (complete loss of eye bristles) is seen when wingless is overexpressed using the sevenless enhancer. That enhancer is easily available from several laboratories and also exists as a GAL4 line. Maybe the authors could use this instead of the GMR GAL4 driver to limit the cells in which the SerT is reduced. It would be interesting if the loss of bristles could be recapitulated by removing SerT in just the sevenless expressing cells.

>Response: Thank you for your valuable suggestions. We require a specific cell type to explore the SerT effect, and your information is really helpful. In this study, with limited time, we cannot investigate this issue; however, future studies are warranted.

Round 2

Reviewer 1 Report

The authors are persuasive in their arguments and the current manuscript is better balanced between the evidence presented and the conclusions. The data clearly indicate a requirement for SerT, but in which cells and how is it not known. Inhibition of apoptosis and Pi3K expression partially rescue.

Specific points that remain: SerT -KD does not produce a characteristic phenotype of reduced PI3K / Akt (small size of the ommatidia). Therefore, one must assume that the reduction of SerT affects not significantly the activity of this pathway or that enough remains for promoting normal ommatidial cell growth. for cell phones and ommatidia to grow to the correct size.

Thus, in my opinion, the authors cannot affirm so strongly that SerT affects the activity of the pathway. I suggest that the authors change line 178 to state “Taken together, the levels of Akt phosphorylation is reduced by SerT knockdown. The levels of pAKT are negatively regulated by a feedback by Akt itself and it is often complex to interpret changes in pAkt. This observation alone is not a proof of reduced Pi3k/Akt, yet it is suggestive.

Figure 4. should therefore indicate the data presented: "The elimination of SerT reduced the levels of phosphorylated Akt protein.

PI3K overexpression supports the role of this pathway but by itself it is not a demonstration that levels are reduced it also shows the capacity of the Pi3K to suppress apoptosis associated with SerT defect.

I still do not see that the authors demonstrate compensatory proliferation.  There are no changes in mitosis, therefore one cannot state that there is an increase in cell proliferation. Compensatory proliferation is to ‘compensate’ and when associated with cell death or cell damage, the dying cells are replaced by those produced by compensatory cell proliferation and the tissue is repaired and the phenotype restored. In my opinion, the authors may be incorrect here by implying that compensatory proliferation somewhat is part of the problem not or the solution. The authors demonstrate convincingly that there is an increased EdU indicating increased number of cells entering S-phase. It is also interesting that Diap1 can rescue also EdU, suggesting that the two phenomena are linked.

I would suggest to rephrase this to propose that SerT-KD causes defects in the cell cycle with dramatic increases in EdU posterior to the second mitotic wave, without changes in mitotic levels, and eliminate the discussion of ‘compensatory proliferation’ hypothesis or include this possibility as an alternative with the caveat that there is no evidence of increased pH3.

It came to my mind, an ‘old’ phenotype that of rux mutants. I revisited an old paper and the authors my want to discuss their results in the light of mutations in the rux gene in their Discussion.

The rux mutation causes eye cells to prematurely enter S phase, giving rise to a rough eye phenotype associated with similar increases in BrdU without changes in mitosis. Rux mutants show a much stronger rough eye than SerT-KD — but SerT is a kd not a null and only in the posterior cells. The defects are strikingly similar.

Here is the citation for rux mutants in the eye: Thomas et al. Cell vol 77, 1003- 1014 (1994).

Author Response

Response to reviewer 1.

Comment 1. Specific points that remain: SerT -KD does not produce a characteristic phenotype of reduced PI3K / Akt (small size of the ommatidia). Therefore, one must assume that the reduction of SerT affects not significantly the activity of this pathway or that enough remains for promoting normal ommatidial cell growth. for cell phones and ommatidia to grow to the correct size.

Thus, in my opinion, the authors cannot affirm so strongly that SerT affects the activity of the pathway. I suggest that the authors change line 178 to state “Taken together, the levels of Akt phosphorylation is reduced by SerT knockdown. The levels of pAKT are negatively regulated by a feedback by Akt itself and it is often complex to interpret changes in pAkt. This observation alone is not a proof of reduced Pi3k/Akt, yet it is suggestive.

>Response: Thank you for your comment. We used your suggested wording to describe this finding (lines 175–179).

Comment 2. Figure 4. should therefore indicate the data presented: "The elimination of SerT reduced the levels of phosphorylated Akt protein.

PI3K overexpression supports the role of this pathway but by itself it is not a demonstration that levels are reduced it also shows the capacity of the Pi3K to suppress apoptosis associated with SerT defect.

> Response: We changed the description of Figure 4, as follows: “Knockdown of SerT reduces levels of phosphorylated Akt protein.”(Line 161).

Comment 3. I still do not see that the authors demonstrate compensatory proliferation.  There are no changes in mitosis, therefore one cannot state that there is an increase in cell proliferation. Compensatory proliferation is to ‘compensate’ and when associated with cell death or cell damage, the dying cells are replaced by those produced by compensatory cell proliferation and the tissue is repaired and the phenotype restored. In my opinion, the authors may be incorrect here by implying that compensatory proliferation somewhat is part of the problem not or the solution. The authors demonstrate convincingly that there is an increased EdU indicating increased number of cells entering S-phase. It is also interesting that Diap1 can rescue also EdU, suggesting that the two phenomena are linked.

I would suggest to rephrase this to propose that SerT-KD causes defects in the cell cycle with dramatic increases in EdU posterior to the second mitotic wave, without changes in mitotic levels, and eliminate the discussion of ‘compensatory proliferation’ hypothesis or include this possibility as an alternative with the caveat that there is no evidence of increased pH3. 

It came to my mind, an ‘old’ phenotype that of rux mutants. I revisited an old paper and the authors my want to discuss their results in the light of mutations in the rux gene in their Discussion.

The rux mutation causes eye cells to prematurely enter S phase, giving rise to a rough eye phenotype associated with similar increases in BrdU without changes in mitosis. Rux mutants show a much stronger rough eye than SerT-KD — but SerT is a kd not a null and only in the posterior cells. The defects are strikingly similar.

Here is the citation for rux mutants in the eye: Thomas et al. Cell vol 77, 1003- 1014 (1994).

> Response: Thank you for your valuable comment. We re-phrased the Results and Discussion sections accordingly: SerT-KD caused defects in the cell cycle and increased the number of S-phase cells but did not affect mitosis (Line 125, 126, 131, 132, 134, and 135).

Additionally, we added a paragraph discussing the rux mutant phenotype in the Discussion section (lines 195–208).

Reviewer 2 Report

Overall I am satisfied by the prompt corrections. I have following suggestions: 

Comment 1: Fig. 1C: I am satisfied with the clarification of authors about the antibody  and the western blot image. However, if authors decides to include Fig. 1C, authors should mention the molecular weight (kilodaltons, eg. 70 kDa) of the bands. Also, flies carry actin5c and not beta-actin. Authors could write only actin. Additionally, authors should include a method section for western blot. Alternatively, authors could omit Fig. 1C entirely. 

Comment 2: I am satisfied with the use of phopho-AKT antibody. Fig. 4E representation of statistics can be changed. Here, the ideal comparison is between GMR GAL4 anterior to GMR>SerT-IR anterior and GMR GAL4 posterior to GMR>SerT-IR posterior and not to GMR>SerT-IR posterior for all data sets.

Grammatical error line 159 and 161: 'the region which IS/GOT affected by SerT-KD' 'the region which did not GET affected by SerT'

Grammatical error line 196 'due to abnormal RECRUITMENT ( instead of recruiting them)'

Author Response

Response to reviewer 2

Comment 1: Fig. 1C: I am satisfied with the clarification of authors about the antibody and the western blot image. However, if authors decides to include Fig. 1C, authors should mention the molecular weight (kilodaltons, eg. 70 kDa) of the bands. Also, flies carry actin5c and not beta-actin. Authors could write only actin. Additionally, authors should include a method section for western blot. Alternatively, authors could omit Fig. 1C entirely. 

> Response: Thank you for your comment. We decided to omit Figure 1C, because we deemed it unnecessary.

Comment 2: I am satisfied with the use of phopho-AKT antibody. Fig. 4E representation of statistics can be changed. Here, the ideal comparison is between GMR GAL4 anterior to GMR>SerT-IR anterior and GMR GAL4 posterior to GMR>SerT-IR posterior and not to GMR>SerT-IR posterior for all data sets.

> Response: We changed the comparison accordingly (Figure 4E).

Grammatical error line 159 and 161: 'the region which IS/GOT affected by SerT-KD' 'the region which did not GET affected by SerT'

>Response: We corrected the grammar in these lines accordingly.

Grammatical error line 196 'due to abnormal RECRUITMENT ( instead of recruiting them).

>Response: We replaced “recruiting them” with “recruitment”, as suggested (line 207).

Reviewer 3 Report

I think that a careful characterization of the developmental defects underlying the rough/bald eye phenotype is essential for the reader to understand the role that SerT is playin in eye development. I really feel that this must be included in the manuscript. For the larval and pupal discs, one can simply use Elav and Cut antibodies to analyze photoreceptor and cone cells. Since the development of the eye is stereotyped and since dozens of papers have analyzed ommatidial assembly in wild type and mutants, the authors should be able to, with relative ease, determine what happens to ommatidial assembly in SerT knockdown mutants. Retinal sections are relatively easy to perform as well. 

While I am cognizant that the covid-19 pandemic is making research difficult, the characterization of ommatidial assembly is essential (in my opinion) and within the authors abilities. 

Author Response

Thank you for your comment. There are a few things we would like to highlight.

Please see the attachment for detailed response.

Round 3

Reviewer 3 Report

I appreciate the willingness of the authors to include the images and text documenting the extra cone cell phenotype. The manuscript is much improved and I recommend that it be published in IJMS. It is an interesting study and will be read by a large number of scientists.